# Correlation between Interleukin-17, High Sensitivity C-Reactive Protein and Pepsinogen in *Helicobacter pylori* Infected Gastritis

Jeanne Winarta [1,*], Bradley Jimmy Waleleng [1], Nelly Tandean Wenas [1], Fujiyanto [2], Oscar Miguna [2] and Marco Rahardja [2]

[1] Division of Gastroenterology and Hepatology, Department of Internal Medicine, Faculty of Medicine, Sam Ratulangi University/Prof. Dr. R. D. Kandou Hospital, Manado 95115, Indonesia; walelengbj@yahoo.com (B.J.W.); insightunsrat1@gmail.com (N.T.W.)

[2] Department of Internal Medicine, Faculty of Medicine, Sam Ratulangi University/Prof. Dr. R. D. Kandou Hospital, Manado 95115, Indonesia; fujiyanto2907@gmail.com (F.); oscar_aloys@hotmail.com (O.M.); rahardjamarco@gmail.com (M.R.)

\* Correspondence: jean_winarta@yahoo.com

**Abstract:** Gastritis is an inflammatory process in the gastric mucosa and submucosa caused by *Helicobacter pylori* (*H. pylori*). The infection modulates immune components, such as interleukin (IL) 17, high sensitivity C-reactive protein (hsCRP) and pepsinogen. This study aimed to determine the relationship between IL-17, hsCRP and pepsinogen in *H. pylori* infected gastritis. This observational cross-sectional study was conducted at Prof. Dr. R. D. Kandou General Hospital Manado from May-July 2022. Measurement of blood sample levels of IL-17, hsCRP, pepsinogen I, pepsinogen II and pepsinogen I/II ratio. Spearman's statistical test was used to determine correlations between these variables. This study involved 48 patients aged 21–64, with a majority of females (67%). IL-7 had a positive correlation with pepsinogen I (r = 0.292; *p* = 0.044) and pepsinogen II (r = 0.288; *p* = 0.047) in *H. pylori* infected gastritis. Meanwhile, IL-17 with pepsinogen I/II ratio, hsCRP with pepsinogen I, pepsinogen II, pepsinogen I/II ratio and IL-17 with hsCRP did not show a significant correlation (*p* > 0.05). There was a correlation between IL 17 to pepsinogen I and pepsinogen II in gastritis infected with *H. pylori*, suggesting the importance of these early markers of inflammation in determining the severity of gastric mucosal inflammation in pylori-infected patients.

**Keywords:** interleukin-17; hsCRP; pepsinogen I; pepsinogen II; pepsinogen ratio I/II; *Helicobacter pylori*

## 1. Introduction

Gastritis is an inflammatory process of the gastric mucosa and submucosa which is mainly caused by infection of *Helicobacter pylori* (*H. pylori*). Other causes of gastritis are bile reflux, non-steroidal anti-inflammatory drugs, autoimmune diseases and viral infections such as mycobacterium avium, herpes simplex and cytomegalovirus. In addition, excessive alcohol consumption and corticosteroid drugs can also cause gastritis [1]. Gastritis is a gastrointestinal problem that often manifests and affects individuals in developing countries with a prevalence of 50.8% [2,3].

*H. pylori* infected ±4.4 billion of the global population in 2015, with a distribution of 24.4–70.1% between continents (Africa vs. Oceania) and 18.9–87.7% between countries (Switzerland vs. Nigeria) [4]. In Indonesia, the prevalence of *H. pylori* infection varies from 10.1% to 48%, especially among ethnic Chinese [5,6]. Sulawesi Island occupies the top 4 position in the prevalence of gastritis due to *H. pylori* infection with a rate of 14.9%. Sulawesi have the highest rates of antibiotic resistance (metronidazole 88.9% and levofloxacin 44.4%) [6]. Particularly in cities in North Sulawesi such as Manado, the prevalence of *H. pylori* almost resembles the national prevalence of 14.34% [7].

This *H. pylori* bacteria causes an infection in the stomach due to its acid-resistant nature, which colonizes and damages the gastric mucosa which causes inflammation, ulcers to perforation [8]. The infection caused by *H. pylori* in the digestive system, especially the stomach, modulates immune components, such as the cytokine family interleukin-17 (IL) and c-reactive protein (CRP). Interleukin-17 is a cytokine that functions in a pro-inflammatory state. IL-17A and IL-17C produced by T-helper 17 (Th17) cells have been associated with protection against microbial infection, where the complex will recruit inflammatory cells into the stomach [9]. Previous studies have found that IL-17 is essential with *H. pylori* infection and severity [10,11]. Study by Kabir et al. in 2011 in Sweden and Contretas et al. in 2015 in America with a population of *H. pylori* infected gastritis patients with an increase of IL-17.10.11 IL-17 was found with 4.5 times higher mRNA expression in gastric biopsy results of *H. pylori* patients. The increase was regulated by the CagA gene ($p = 0.012$), I$\kappa$B kinase, extracellular mitogen signal-regulated kinase and Jun N-terminal kinase [12].

CRP levels also have an important role in the incidence of gastritis due to *H. pylori* infection. Jafarzadeh et al. found that CRP levels and high sensitivity CRP (hsCRP) increased with the incidence and severity of *H. pylori* gastritis ($p$ value < 0.04) [13]. The use of hsCRP is more applicable than CRP because of its ability to detect subclinical inflammatory status reflects the presence of an inflammatory process even though the CRP is still low [14].

On the other hand, pepsinogen is a gastric mucosal biomarker which is predominantly secreted into the gastric lumen [15]. Type I pepsinogen is mostly secreted by mucosal cells located in the fundus of the stomach., whereas type II pepsinogen by chief cells in the proximal duodenal mucosa and throughout the stomach [16]. Increased pepsinogen concentration I and II can occur due to the inflammatory process and can increase in gastritis. When gastritis occurs, gland dysfunction will occur which will affect type I or II pepsinogen and their ratio. However, pepsinogen I is more easily affected leading to a decrease in the pepsinogen I/II ratio [15]. Yoshida et al. found a difference in pepsinogen I/II ratio 3.0, pepsinogen II 10.4 and pepsinogen I 3.8 ng/mL in patients with *H. pylori* infection compared to those without *H. pylori* infection [17]. Pepsinogen I/II ratio in *H. pylori* infection was found to be influenced by external factors such as race, age, gender and malignancy [18]. Although the specific role of pepsinogen has not been widely studied, the pepsinogen I/II ratio has adequate ability to assess *H. pylori* eradication [19].

Based on what has been described above, further research is needed to assess the relationship between IL-17, hsCRP with pepsinogen I, pepsinogen II and pepsinogen I/II ratio in gastritis patients infected with *H. pylori*. Until now there has been no research on the relationship of IL-17, hsCRP, pepsinogen I, pepsinogen II and their ratio in gastritis patients infected with *H. pylori*, especially in Indonesia. This study aimed to assess the relationship of the inflammatory factors in *H. pylori* infected gastritis with pepsinogen I, pepsinogen II and pepsinogen I/II ratio.

## 2. Materials and Methods

### 2.1. Study Design

We conducted an analytic observational study with a cross-sectional study design. This study was conducted at the gastroenterology polyclinic and the Internal Medicine ward of Prof. Dr. R. D. Kandou Hospital Manado. The esophagogastroduo-denoscopy procedure was carried out at the gastro-intestinal endoscopy center in Prof. Dr. R. D. Kandou Hospital Manado. Measurement of blood sample levels of IL-17, hsCRP, pepsinogen I, pepsinogen II and pepsinogen ratio I/II were carried out at the Prodia Laboratory Manado. Overall, the study was conducted from May–July 2022. This study has been approved by the Research Ethics Committee of Prof. Dr. R. D. Kandou Hospital.

## 2.2. Patient Selection and Data Collection

In this study, consecutive sampling was performed. The study involved patients aged ≥18 years with a diagnosis of gastritis due to *H. pylori* infection based on a rapid urease test and histopathologic results of endoscopic biopsy and willing to participate in this study. Patients with malignancy, acute infection, autoimmune disease, who were pregnant, consuming alcohol regularly, taking NSAIDs in the previous 14 days, having pulmonary tuberculosis infection, taking proton pump inhibitor (PPI)/H2 receptor antagonist (ARH2) drugs for 2 weeks before the examination, or using cholinergic/anticholinergic drugs, were excluded from the study. Data pertaining to anamnesis, physical examination, patient's medical record and supporting investigation tests were collected.

Diagnosis of gastritis due to H. pylori infection was based on campylobacter-like organism (CLO) test performed with AMA® Rapid Urease Test (RUT) examination and confirmed with pathology results. The CLO test was performed during the esophagogastroduodenoscopy (EGD) and a biopsy of tissue from the gastric corpus and antrum was performed. The tissue was placed into a medium containing pH indicators such as phenol red. Urease from H. pylori hydrolyzed urea to ammonia, which increased the pH of the medium.

Interleukin-17 (IL-17) cytokine levels were measured using a serum Enzyme-Linked Immunisirbent Assay (ELISA) with kit ab100665. Sterile 96-well plate coated with coating antibodies on the first day, incubated at 4–8 °C overnight, washed with 20 L of phosphate-buffered saline (PBS) solution twice (pH 7.4), added incubation buffer (PBS, 0.05% Tween 20 [polyoxyethylene sorbitol ester], 1% bovine serum albumin [BSA]) 200 L, incubated at room temperature for 1 h and washed five times with wash buffer. ELISA absorbance was measured at 450 nm using the ELISA MPSCREEN MR-96A reader.

The hsCRP test was measured using standard laboratory methods of intravenous blood in each patient diagnosed with gastritis due to H. pylori infection. Serum pepsinogen I and II was measured by obtaining five mL of blood samples from cubital vein in resting conditions after fasting overnight. The commercial ELISA assay (Gastropanel®, BioKit®, Helsinski, Finland) following the manufacturer's instructions was used to measure pepsinogen I and II levels.

## 2.3. Statistical Analysis

Data analysis was performed using the Statistical Package for the Social Sciences software (version 26.0; IBM, Chicago, IL, USA). Univariately, data variables such as age, gender, IL-17, hsCRP, pepsinogen I, pepsinogen II and pepsinogen I/II ratio will be described in terms of frequency, percentage, mean, standard deviation (SD), median, minimum and maximum. Data normality will be analysed with Shapiro–Wilk. The data distribution is normal if the $p$ value > 0.05.

Correlation analysis is used to find the relationship between variables using the Spearman correlation test if the data is not normally distributed or Pearson test if the data is normally distributed. Data will also be included the strength of the correlation (r) and the direction of the correlation whether positive or negative. Correlation interpretations are: 0–0.2 (very weak), 0.21–0.4 (weak), 0.41–0.6 (fair), 0.61–0.8 (strong) and 0.81–1 (very strong). $p$ values < 0.05 and 95% confidence intervals are also included. All results will be presented in the form of scatter plots or tables and assisted with narratives for explanation.

## 3. Results

### 3.1. Characteristics of Study Populations

This research was carried out from May to July 2022 at the gastroenterology polyclinic, RSUP Prof. Dr. R. D. Kandou Manado. In this study, there were 48 patients involved, consisting of 16 males and 32 females, with a median age of 42 (21–64) years. The characteristics of the sample can be seen in Table 1.

**Table 1.** Characteristics of study variables.

| Characteristics | N | Mean + SD | Median |
|---|---|---|---|
| Age (Year) | 48 | 41.98 ± 11.60 | 4.50 (21.00–64.00) |
| Gender | | - | - |
| Male | 16 | | |
| Female | 32 | | |
| IL-17 (mg/dL) | 48 | 2.95 ± 1.47 | 3.10 (0.20–8.50) |
| hs CRP (mg/dL) | 48 | 3.91 ± 6.87 | 1.50 (0.20–29.90) |
| Pepsinogen I (mcg/L) | 48 | 200.15 ± 92.83 | 187.15 (89.80–467.40) |
| Pepsinogen II (mcg/L) | 48 | 18.07 ± 10.83 | 14.71 (4.24–45.42) |
| Pepsinogen I and II ratio (mcg/L) | 48 | 12.84 ± 5.31 | 12.28 (5.39–38.89) |

*3.2. Correlation of IL-17 and Pepsinogen I in H. pylori Infected Gastritis*

The data normality test with the Shapiro–Wilks test found that the data distribution was not normal, so to determine the correlation between IL-17 and pepsinogen I in gastritis infected with *H. pylori*, the Spearman correlation test was performed (Table 2). The results in Table 2 show that there is a significant positive correlation between IL-17 and pepsinogen I in gastritis patients infected with *H. pylori* and the correlation is weak. This indicates that the higher the IL-17, the higher the level of pepsinogen I in gastritis patients infected with *H. pylori*.

**Table 2.** Correlation of IL-17 and Pepsinogen I in *H. pylori* infected gastritis.

| Variable Correlation | N | Correlation Coefficient (r) | *p* Value |
|---|---|---|---|
| IL-17—Pepsinogen I in Gastritis patient infected with *H. pylori* | 48 | 0.292 | 0.044 |

*3.3. Correlation of IL-17 and Pepsinogen II in H. pylori Infected Gastritis*

The data normality test using the Shapiro–Wilks test found that the data distribution was not normal, so to determine the correlation between IL-17 and pepsinogen II in gastritis infected with *H. pylori*, the Spearman correlation test was performed (Table 3). The results in Table 3 show that there is a significant positive correlation between IL-17 and pepsinogen II in gastritis patients infected with *H. pylori*, and the correlation is weak. This indicates that the higher the IL-17, the higher the level of pepsinogen II in gastritis patients infected with *H. pylori*.

**Table 3.** Correlation of IL-17 and Pepsinogen II in *H. pylori* infected gastritis.

| Variable Correlation | N | Correlation Coefficient (r) | *p* Value |
|---|---|---|---|
| IL-17—Pepsinogen II in Gastritis patient infected with *H. pylori* | 48 | 0.288 | 0.047 |

*3.4. Correlation of IL-17 and Pepsinogen I/II Ratio in H. pylori Infected Gastritis*

The data normality test using the Shapiro–Wilks test found that the data distribution was not normal, so to determine the correlation between IL-17 and pepsinogen I/II ratio in gastritis infected with *H. pylori*, the Spearman correlation test was performed (Table 4). The results in Table 4 show that there is an insignificant negative correlation between IL-17 and pepsinogen I/II ratio in gastritis infected with *H. pylori*, and the correlation is very weak. This indicates that the higher the IL-17, the lower the pepsinogen I/II ratio in gastritis patients infected with *H. pylori*.

**Table 4.** Correlation of IL-17 and Pepsinogen I/II Ratio in *H. pylori*-infected Gastritis.

| Variable Correlation | N | Correlation Coefficient (r) | *p* Value |
|---|---|---|---|
| IL-17—Pepsinogen I//II ratio in Gastritis patient infected with *H. pylori* | 48 | −0.138 | 0.350 |

### 3.5. Correlation of hsCRP and Pepsinogen I in H. pylori Infected Gastritis

The data normality test using the Shapiro–Wilks test found that the data distribution was not normal, so to determine the correlation between hsCRP and pepsinogen I in gastritis infected with *H. pylori*, the Spearman correlation test was performed (Table 5). The results in Table 5 show that there is an insignificant negative correlation between hsCRP and pepsinogen I in gastritis patients infected with *H. pylori*, and the correlation is very weak. This indicates that the higher the hsCRP, the lower the pepsinogen I level in gastritis patients infected with *H. pylori*.

**Table 5.** Correlation of hsCRP and Pepsinogen I in *H. pylori* infected gastritis.

| Variable Correlation | N | Correlation Coefficient (r) | *p* Value |
|---|---|---|---|
| hsCRP—Pepsinogen I in Gastritis patient infected with *H. pylori* | 48 | −0.043 | 0.771 |

### 3.6. Correlation of hsCRP and Pepsinogen II in H. pylori Infected Gastritis

The data normality test using the Shapiro–Wilks test found that the data distribution was not normal, so to determine the correlation between hsCRP and pepsinogen II in gastritis infected with *H. pylori*, the Spearman correlation test was performed (Table 6). The results in Table 6 show that there is an insignificant negative correlation between hsCRP and pepsinogen II in gastritis patients infected with *H. pylori*, and the correlation is very weak. This indicates that the higher the hsCRP, the lower the pepsinogen II level in gastritis patients infected with *H. pylori*.

**Table 6.** Correlation of hsCRP and Pepsinogen II in *H. pylori* infected gastritis.

| Variable Correlation | N | Correlation Coefficient (r) | *p* Value |
|---|---|---|---|
| hsCRP—Pepsinogen II in Gastritis patient infected with *H. pylori* | 48 | −0.070 | 0.636 |

### 3.7. Correlation between hsCRP and Pepsinogen I/II Ratio in H. pylori-Infected Gastritis

The data normality test using the Shapiro–Wilks test found that the data distribution was not normal, so to determine the correlation between hsCRP and pepsinogen ratio I/II in gastritis infected with *H. pylori*, the Spearman correlation test was performed (Table 7). The results in Table 7 show that there is an insignificant positive correlation between hsCRP and pepsinogen I/II ratio in gastritis patients infected with *H. pylori* and the correlation is very weak. This indicates that the higher the hsCRP, the higher the pepsinogen I/II ratio in gastritis patients infected with *H. pylori*.

**Table 7.** Correlation of hsCRP and Pepsinogen I/II ratio in *H. pylori* infected gastritis.

| Variable Correlation | N | Correlation Coefficient (r) | *p* Value |
|---|---|---|---|
| hsCRP—Pepsinogen I/II ratio in Gastritis patient infected with *H. pylori* | 48 | −0.095 | 0.523 |

### 3.8. Correlation of IL-17 and hsCRP in H. pylori Infected Gastritis

The data normality test with the Shapiro–Wilks test found that the data distribution was not normal, so to determine the correlation between IL-17 and hsCRP in gastritis

patients infected with *H. pylori*, the Spearman correlation test was performed (Table 8). The results in Table 8 show that there is an insignificant negative correlation between IL-17 and hsCRP in gastritis infected with *H. pylori*, and the correlation is very weak. This indicates that the higher the IL-17, the lower the hsCRP levels in gastritis patients infected with *H. pylori*.

**Table 8.** Correlation of IL-17 and hsCRP in *H. pylori* infected gastritis.

| Variable Correlation | N | Correlation Coefficient (r) | *p* Value |
|---|---|---|---|
| IL-17—hsCRP in Gastritis patient infected with *H. pylori* | 48 | −0.062 | 0.673 |

## 4. Discussion

The results showed that most of patient in this study were female (66.6%). This characteristic was also found in a study by Bordin and Plavnik in Russia using the 3 C-urea breath tests. A higher prevalence was found in female than male subjects (57.6% vs. 52.6%) [20,21]. This result was in accordance with the 2019 study of Hong et al. in China. The study with 53,260 subjects showed that the prevalence of *H. pylori* infection was higher in women than men in all age categories (49.2% vs. 47.9%), *p* = 0.002 [22].

Several international literatures discuss the age characteristics on the incidence of *H. pylori* infection. In this study, it was found that the average age in this study was 41.98 years, with the median 42.5 years, where the youngest age in this study was 21 years, while the oldest age was 64 years. Age is mentioned as one of the variables that has been widely studied in epidemiological studies of *H. pylori* infection. The study by Husein et al. (2008) showed the prevalence of *H. pylori* infection increased with age with a maximum of 85% in adult subjects aged 40–60 years [13]. This statement was also supported by Khalife et al., 2017 in an epidemiological study in Lebanon. Epidemiological studies showed the highest prevalence of *H. pylori* in the age group of 51–80 (69.81%) and the lowest prevalence in the age group of 1–20 years (17.3%) and found a significant relationship between the two groups with *H. pylori* infection. ($x^2$ = 28.08, *p* value < 0.0001) [14].

Interleukin 17 is a cytokine compound that plays an important role in the course of *H. pylori* infection. In subjects with normal IL-17 levels, a significant increase occurred after being exposed to *H. pylori* infection [23]. An increase in IL-17 along with other immune cells produced by Th-17 cells correlated with the severity of infection of *H. pylori*.

The correlation between IL-17 and pepsinogen I showed a significantly weak positive correlation (r = 0.292; *p* = 0.044). This correlation is stated as a weak correlation but can be used for further research considering the significant *p* value (*p* < 0.05). As with pepsinogen I, the results in this study also showed a positive correlation between IL-17 and pepsinogen II in subjects with *H. pylori*-infected gastritis (r = 0.288; *p* < 0.047). This correlation is weak positive but can be considered, given its strong significance value (*p* < 0.05).

Interleukin 17 in conjunction with IL-8 may be involved in inducing gastric ulcers because IL-8 contributes to neutrophil recruitment at the ulcer site. According to signaling pathway, IL-17 has been shown to activate ERK 1/2 MAP kinase in gastric epithelial cell lines and in gastric epithelial cells that have been isolated from *H. pylori* [24]. The mechanism of the inflammatory pathway of the *H. pylori* pathway will affect other inflammatory factors such as IL-16, IL-8, and TNF alpha which causes an increase in gastrin gastric and which leads to an increase in pepsinogen production. At the beginning of the initial phase of gastric mucosal inflammation caused by *H. pylori* infection, there is an increase in pepsinogen I and pepsinogen II, but when inflammation progresses to atrophic gastritis, there will be a decrease in pepsinogen I < 70 mcg/L and pepsinogen I/II ratio < 3. It can be seen in this study that atrophic gastritis has not occurred with insignificant pepsinogen ratio I and II. The correlation of IL-17 to pepsinogen I and pepsinogen II is a weak positive correlation because the increase in IL-17 and pepsinogen is quite sensitive to *H. pylori*

infection but is not specific. Until the time this research was conducted, an analysis of the relationship between IL-17 and pepsinogen I/II had not been conducted.

In this study there was an increase in pepsinogen I levels but no increase in pepsinogen II. This result can be caused by different normal values in various populations [25–27]. The results of the study in China by Zhang et al. showed that serum pepsinogen level was influenced by demographic factors including gender, age, smoking, drinking alcohol and dietary habits, serum pepsinogen test methodologies such as radioimmunoassay or enzyme immunoassay, which explained varied serum pepsinogen limit levels in different populations. Previous studies have also shown that serum pepsinogen I levels and pepsinogen I/II ratios are useful indicators of gastric mucosal alterations. The sensitivity and specificity of the pepsinogen test differ in various regions and populations. Further researches are needed to improve the accuracy of the pepsinogen test for the diagnosis of gastritis [25,28]. For example, in Japan, the suggested cut off for determining gastric atrophy and risk of gastric cancer is 70 g/L for pepsinogen I and 3.0 for pepsinogen I/II ratio. In European countries, the cut off value is 25 g/L for pepsinogen I and 3.0 for pepsinogen I/II [29]. In a Korean study, pepsinogen I 70 ng/mL showed adequate sensitivity (72.4%) but low specificity. (20.2%), and the sensitivity and specificity of the pepsinogen I/II cut-off ratio 3 were 59.2–61.7% and 61.0%, respectively. Furthermore, different assay technique for serum pepsinogen levels is widely utilized in various regions of the world. For example, the ELISA is mostly used in Europe whereas the latex agglutination test is widely used in Japan [26,29]. The use of different testing methods may lead to potential differences in serum pepsinogen levels. The mean values of pepsinogen II in the healthy Chinese population were 7 microg/L and 6 microg/L for both males and females. Pepsinogen II was significantly increased in gastritis infected with *H. pylori*, and the best limit value of pepsinogen II was 8.25 microg/L. Additionally, Chinese patients with pepsinogen II > 10.25 microg/L had a higher risk of several *H. pylori*-associated gastropathies [26]. In this study, the cut-off value for pepsinogen II was 3–19 mcg/L [30]. The differences in cut off value is the reason why pepsinogen is not significant in this study. The difference in normal values in various populations and different testing methods is the reason for the difference in research results.

Several studies have explained the correlation between pepsinogen levels and *H. pylori* infection. Some studies have reported serum pepsinogen II levels to be higher in *H. pylori* infection and serum pepsinogen II levels have been recommended for use in diagnosing and evaluating *H. pylori* eradication. In contrast, other studies have reported an increase in serum pepsinogen I compared to pepsinogen II in *H. pylori* infection. This study showed there was no significant change in pepsinogen I, pepsinogen II or pepsinogen I/II ratio in patients with *H. pylori* gastritis. It is well established that *H. pylori* is capable of inducing an increase in pepsinogen production during the early stages of mucosal inflammation. which then caused chronic infection to develop into atrophy [26]. In this study, none of the research subjects had chronic gastritis. Lipopolysaccharides contained in the *H. pylori* membrane may be a contributing factor in this infection. This is supported by Young et al. who showed that purified *H. pylori* lipopolysaccharide increased pepsinogen secretion up to 50-fold. Other IL-17-related cytokines are IL-1β, IL-6, IL-21 and IL-23. IL-17/IL-21 and IL-17/IL-23 have antimicrobial and even some inflammatory and hematopoietic effects on epithelial, endothelial and fibroblast cells [31].

High sensitivity C-reactive protein is a marker of inflammation which is mainly caused by bacteria. In gastritis, cytokines such as IL-8 and IL-6 will increase as an inflammatory process. These compounds will circulate and trigger cells to secrete higher levels of hsCRP. In this study, it was shown that hsCRP had a very weak negative correlation with pepsinogen I (r = −0.043; *p* = 0.771), but this value was not significant, as well as pepsinogen II (r = −0.070; *p* = 0.636). Correlation test has also shown that the relationship between hsCRP and pepsinogen ratio I/II shows a very weak and not significant correlation (r = 0.095; *p* = 0.523). This is in accordance with a previous study by Liu H et al. where there was no significant difference in hsCRP levels of *H. pylori* positive and *H. pylori* negative patients (*p* > 0.05) [32].

Several studies have shown that hsCRP production is controlled by proinflammatory cytokines, such as TNF, IL-1 and IL6. Elevated IL-1 and IL-6 have been found in individuals with *H. pylori* infection, especially in atrophic gastritis. In this study, no atrophic gastritis was found in the research subjects.

Significant differences in hsCRP between *H. pylori* and non-*H. pylori* gastritis have also been noted in studies [33], although the mechanism for the synthesis of hsCRP in *H. pylori* infection is still unclear. However, several studies have shown that hsCRP production is regulated by proinflammatory cytokines, such as TNF, IL-1 and IL-6. Elevated levels of IL-1 and IL-6 have been found in individuals with *H. pylori* infection. The proinflammatory cytokines can stimulate the hsCRP production during *H. pylori* infection [13]. The hsCRP level in patients with a pepsinogen I/II ratio < 3 is higher than in patients with a pepsinogen I/II ratio > 3. The correlation between hsCRP and pepsinogen I/II ratio has been demonstrated in a study by Liu et al. in 2020. The results of this study showed a positive correlation of hsCRP, pepsinogen I, pepsinogen II and pepsinogen I/II ratio with age (r = 0.210, 0.287 and 0.133; $p < 0.05$). In gastritis, cytokines such as IL-8 and IL-6 will increase in as an inflammatory process. These cytokines will circulate and trigger cells to secrete higher levels of hsCRP, particularly in atrophic gastritis patients. The study also found that serum hsCRP levels were influenced by the severity of acute and chronic inflammation [20,23]. In this study, patients with chronic gastritis were found in the initial phase of early inflammation, thus showing an insignificant relationship between hsCRP with pepsinogen and IL17. Another factor that may contribute to this hypothesis is the presence of genetic factors CagA and VagA in the inflammatory process. In addition, the number of colonization bacterial density also plays a role in the process of inflammation.

Interleukin 17 is a crucial component of the immune system. IL-17 levels in the gastric mucosal of antrum were elevated in patients infected with *H. pylori*, particularly in the chronic phase [26,34]. The levels of hsCRP were also found to be markedly elevated in gastritis patients with H.pylori infection. Although at present the mechanism for the synthesis of hsCRP in *H. pylori* infection has not been clearly identified, an increase in proinflammatory cytokines such as IL-1 and IL-6 is thought to stimulate the production of hsCRP in *H. pylori* infection. The correlation test in this study found a very weak negative correlation between IL-17 and hsCRP in gastritis patients with *H. pylori* infection (r= −0.062; $p = 0.673$), but the correlation is insignificant ($p > 0.05$). The correlation study of IL-17 and hsCRP has never been conducted. Further studies are needed to compare the levels of IL-17, hsCRP and pepsinogen in the control group [27].

Our study has limitations. The degree of gastritis was not assessed in this study. Hence, further research is needed to assess the degree of gastritis, making it easier for clinicians to evaluate the effect of inflammatory factors, such as IL-17 and hsCRP to pepsinogen I, pepsinogen II and pepsinogen I/11 ratios in patients with gastritis due to H. pylori infection. The presence of inflammatory markers could help clinicians in screening and determining treatment for eradication to suppress the progression of gastric mucosal damage.

## 5. Conclusions

There was a significant relationship between IL-17 and pepsinogen I and pepsinogen II, but there was no significant relationship between IL-17 and pepsinogen I/II ratio in gastritis infected with *H. pylori*. There was no significant relationship between hs-CRP with pepsinogen I, pepsinogen II and pepsinogen I/II ratio. This is presumed to be due to the study subjects having chronic gastritis that had not or did not develop until the occurrence of atrophic gastritis.

**Author Contributions:** J.W., B.J.W. and N.T.W. contributed to the conception and design of the work. J.W., B.J.W., N.T.W., F., O.M. and M.R. conducted the work, collected and analysed the data. J.W., F., O.M. and M.R. drafted the manuscript. J.W., B.J.W. and N.T.W. reviewed and revised the manuscript critically. All authors have read and agreed to the published version of the manuscript.

**Funding:** This study received no external funding.

**Institutional Review Board Statement:** The study was conducted in accordance with the Declaration of Helsinki, and approved by the Institutional Review Board (or Ethics Committee) of General hospital Prof. Dr. R. D. Kandou Hospital (protocol code: 0.98/EC/UEPK-KANDOU/VI/2022 and date of approval: 13 June 2022) for studies involving humans.

**Informed Consent Statement:** Inform consent was obtained from all subjects involved in the study.

**Data Availability Statement:** The data used to support the findings of this study are available from the corresponding author upon request.

**Acknowledgments:** The authors would like to thank the medical staff of Gastroenterology Polyclinic and Internal Medicine Ward at Prof. Dr. R. D. Kandou Hospital Manado. The authors also thank to the participants who participated in this study.

**Conflicts of Interest:** The authors declare no conflicts of interest.

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
