# Peer review of "Correlation between Interleukin-17, High Sensitivity C-Reactive Protein and Pepsinogen in Helicobacter pylori Infected Gastritis"

_gastroent, doi:10.3390/gastroent15010003_

Round 1
Reviewer 1 Report
Comments and Suggestions for Authors
This manuscript shows that there was a correlation between IL-17 to pepsinogen I and pepsinogen II in gastritis infected with H. pylori, suggesting the importance of these early markers of inflammation in determining the severity of gastric mucosal inflammation in pylori-infected patients.
The methodology and the statistical analyzes don't have any problems.
The discussion is well considered.
So I think this manuscript should be accepted.
Author Response
Dear Reviewer,
Thank you for your valuable comments and suggestions. We really appreciate you and the reviewers for your precious time in reviewing our paper. It was your valuable and insightful comments that led to possible improvements in the current version. The authors have carefully considered the comments and tried our best to address every one of them. We hope the manuscript after careful revisions meet your high standards. The authors welcome further constructive comments, if any.
Sincerely,
Authors
Reviewer 2 Report
Comments and Suggestions for Authors The article is well designed and I think it would be appropriate to publish it.Author Response
Dear Reviewer,
Thank you for your valuable comments and suggestions. We really appreciate you and the reviewers for your precious time in reviewing our paper. It was your valuable and insightful comments that led to possible improvements in the current version. The authors have carefully considered the comments and tried our best to address every one of them. We hope the manuscript after careful revisions meet your high standards. The authors welcome further constructive comments, if any.
Sincerely,
Authors
Reviewer 3 Report
Comments and Suggestions for Authors
- This is original submission; Therefore, abstract should be contains background, method, results and conclusion.
- Inclusion and exclusion criteria should be suggested.
- The author should determined how H. pylori infection was confirmed?
- The details of IL-17, hsCRP, pepsinogen I, pepsinogen II assessment should be specified.
- Sample size was low??
- Discuss about the novelty of this work.
- Discuss about study limitation.
- the manuscript could benefit from figures.
- Conclusion should be objective with further perspectives.
Comments on the Quality of English LanguageThe manuscript need minor polishing.
Author Response
r Reviewer,
Thank you for your comments and suggestions. The manuscript has been revised, which is highlighted. We would like to provide you the details of the revision to the manuscript and our responses to your comments as below:
- This is original submission; Therefore, abstract should be contains background, method, results and conclusion.
- The abstract already contained background, method, results and conclusion. Hence, no revision is made.
- Inclusion and exclusion criteria should be suggested.
- Inclusion and exclusion criteria have been stated in the method sub-part. Hence, no revision is made.
- The author should determined how H. pylori infection was confirmed?
-
Diagnosis of gastritis due to H. pylori infection was based on campylobacter-like organism (CLO) test performed with AMA®️ Rapid Urease Test (RUT) examination and confirmed with pathology results. The CLO test was performed during the esophagogastroduodenoscopy (EGD) and a biopsy of tissue from the gastric corpus and antrum was performed. The tissue was placed into a medium containing pH indicators such as phenol red. Urease from H. pylori hydrolyzed urea to ammonia, which increased the pH of the medium.
-
- The details of IL-17, hsCRP, pepsinogen I, pepsinogen II assessment should be specified.
-
Interleukin-17 (IL-17) cytokine levels were measured using a serum Enzyme-Linked Immunisirbent Assay (ELISA) with kit ab100665. Sterile 96-well plate coated with coating antibodies on the first day, incubated at 4-8°C overnight, washed with 20 L of phosphate buffer saline (PBS) solution twice (pH 7.4), added incubation buffer (PBS, 0.05% Tween 20 [polyoxyethylene sorbitol ester], 1% bovine serum albumin [BSA]) 200 L, incubated at room temperature for 1 hour, and washed five times with wash buffer. ELISA absorbance was measured at 450 nm using the ELISA MPSCREEN MR-96A reader.
The hsCRP test was measured using standard laboratory methods of intravenous blood in each patients diagnosed with gastritis due to H. pylori infection. Serum pepsinogen I and II was measured by obtaining five mL of blood samples from cubital vein in resting conditions after fasting overnight. The commercial ELISA assay (Gastropanel®️, BioKit®️, Helsinski, Finland) following the manufacturer's instructions was used to measure pepsinogen I and II levels.
-
- Sample size was low??
- This study used correlative analytical design. Using the sample size formula, a minimum sample number of 47 subjects was obtained.
- Discuss about the novelty of this work.
- There has been no research on the relationship of IL-17, hsCRP, pepsinogen I, pepsinogen II, and their ratio in gastritis patients infected with H. pylori, especially in Indonesia
- Discuss about study limitation.
- The degree of gastritis was not assessed in this study. Hence, further research is needed to assess the degree of gastritis, making it easier for clinicians to evaluate the effect of inflammatory factors, such as IL-17 and hsCRP to pepsinogen I, pepsinogen II, and pepsinogen I/11 ratios in patients with gastritis due to H. pylori infection. The presence of inflammatory markers could help clinicians in screening and determining treatment for eradication to suppress the progression of gastric mucosal damage.
- Conclusion should be objective with further perspectives.
- This is presumed to be due to the study subjects had chronic gastritis that had not or did not develop until the occurrence of atrophic gastritis.
Round 2
Reviewer 3 Report
Comments and Suggestions for Authors
Well revised.
Comments on the Quality of English LanguageMinor polishing could be improve the quality of manuscript.